# Do Role-Playing Agents Practice What They Preach? Belief-Behavior Consistency in LLM-Based Simulations of Human Trust

**Amogh Mannekote[1]  Adam Davies[2]  Guohao Li[3]  Kristy Elizabeth Boyer[1]**
**ChengXiang Zhai[2]  Bonnie J Dorr[1]  Francesco Pinto[4]**

[1]University of Florida   [2]University of Illinois Urbana–Champaign   [3]`CAMEL-AI.org`
[4]University of Chicago

## Abstract

As large language models (LLMs) are increasingly studied as role-playing agents to generate synthetic data for human behavioral research, ensuring that their outputs remain coherent with their assigned roles has become a critical concern. In this paper, we investigate how consistently LLM-based role-playing agents' *stated beliefs* about the behavior of the people they are asked to role-play ("what they say") correspond to their *actual behavior* during role-play ("how they act"). Specifically, we establish an evaluation framework to rigorously measure how well beliefs obtained by prompting the model can predict simulation outcomes in advance. Using an augmented version of the GENAGENTS persona bank and the Trust Game (a standard economic game used to quantify players' trust and reciprocity), we introduce a *belief-behavior consistency* metric to systematically investigate how it is affected by factors such as: (1) the types of beliefs we elicit from LLMs, like expected outcomes of simulations versus task-relevant attributes of individual characters LLMs are asked to simulate; (2) when and how we present LLMs with relevant information about Trust Game; and (3) how far into the future we ask the model to forecast its actions. We also explore how feasible it is to impose a researcher's own theoretical priors in the event that the originally elicited beliefs are misaligned with research objectives. Our results reveal systematic inconsistencies between LLMs' stated (or imposed) beliefs and the outcomes of their role-playing simulation, at both an individual- and population-level. Specifically, we find that, even when models appear to encode plausible beliefs, they may fail to apply them in a consistent way. These findings highlight the need to identify how and when LLMs' stated beliefs align with their simulated behavior, allowing researchers to use LLM-based agents appropriately in behavioral studies.

## 1  Introduction

Role-playing agents based on large language models are increasingly used to generate synthetic datasets of human behavior to reduce the high cost of running human subject studies (Park et al., 2023; Hou et al., 2025; Huang et al., 2024; Balog & Zhai, 2025). As the promise of using role-playing agents for performing scientific research and informing policy decision-making gains traction (Sarstedt et al., 2024), it is imperative to rigorously assess their validity to safeguard against flawed inferences and ensure the reliability of ensuing conclusions (Rossi et al., 2024; Agnew et al., 2024).

However, existing evaluation frameworks for LLM-based role-playing agents are inherently post-hoc: they assess an agent's behavior only after the simulation is complete (Argyle et al., 2023; Huang et al., 2024; Wang et al., 2023b; Bhandari et al., 2025; Hou et al., 2025). For instance, researchers compare generated survey responses against human data (Argyle et al., 2023), align self-reported beliefs with open-ended outputs (Huang et al., 2024), evaluate personality traits via dialogue cues (Wang et al., 2023b; Bhandari et al., 2025), or examine

emergent population dynamics through post-simulation evaluation (Hou et al., 2025). Because these methods cannot assess belief-behavior consistency, the consistency between an LLM's elicited beliefs and its subsequent actions during role-play is discovered only after considerable resources have been invested in generating synthetic data, and existing work typically considers either population-level or individual-level evaluation, but not both.

Closely tied to the issue of evaluating a model's belief-behavior consistency is the question of what to do when its elicited beliefs diverge from the researcher's expectations—specifically, whether we can control the simulation by imposing a desired belief. For example, if an LLM continues to portray younger individuals as more generous even after being explicitly instructed to simulate a population in which older adults are more generous, this would raise concerns about the *controllability* of role-playing agents (Shen et al., 2025; Mannekote et al., 2025).

We present a framework that elicits a model's beliefs through targeted prompts to measure belief-behavior consistency in role-play simulations at two levels of analysis. First, at the *population level*, we quantify consistency by computing the correlation between persona attributes and simulated statistical behaviors aggregated across all simulated participants. Second, at the *individual level*, we test an LLM's capacity to predict the future actions of a specific simulated member of the population. In both cases, we test whether querying the model's own expectations can flag misaligned beliefs before they lead to errors in large-scale synthetic data. We also examine three design choices: how much background context we give the model when eliciting beliefs, which outcomes we ask it to predict, and how far into the future we ask it to forecast its actions—and how each choice affects belief-behavior consistency.

We illustrate our general framework with a Trust Game case study (Berg et al., 1995), a standard benchmark for LLM role-playing biases (Wei et al., 2024; Xie et al., 2024). The Trust Game offers quantifies interpersonal trust as the amount of money the first player (the Trustor) chooses to send to the second player (the Trustee). We elicit the model's beliefs about how individuals with specific personas or populations with shared characteristics would act, then have the model role-play as the Trustor, allowing direct comparison of stated beliefs with actual behaviors. Our findings suggest systematic belief-behavior inconsistencies: explicit task context during belief elicitation does not appear to improve consistency, self-conditioning enhances alignment in some models while imposed priors tend to undermine it, and individual-level forecasting accuracy tends to degrade over longer horizons.

Our contributions are the following: (1) We introduce an evaluation framework that uses prompt-based belief elicitation to identify issues with the validity of LLM-based role-playing agents prior to large-scale synthetic data generation. (2) At the population level, we analyze belief-behavior consistency in the Trust Game (Section 4) and assess how belief conditioning—through self-conditioning and the imposition of researcher-defined priors—affects model controllability and consistency (Section 4.3). (3) At the individual level, we evaluate whether LLM agents can reliably forecast their own simulated actions across multi-round Trust Game scenarios (Section 5).

## 2 Related Work

LLM-based role-playing agents have gained prominence as tools for generating synthetic behavioral data for a diverse set of applications (Wang et al., 2024a; Mannekote et al., 2025; Shao et al., 2023; Louie et al., 2024; Wang et al., 2024b): for instance developing interactive characters for open-world games (Yan et al., 2023) to predicting vaccine hesitancy in human populations (Hou et al., 2025). Existing approaches to evaluating these agents involves comparing the agent's outputs to human-generated data.

For instance, Argyle et al. (2023) assess whether survey response patterns generated by an LLM match that of real-world surveys. Huang et al. (2024) create TRUSTSIM, which asks the agent to self-report its beliefs and then checks if its free-form responses align with those beliefs. Other works, such as Wang et al. (2023b) and Bhandari et al. (2025), focus on whether an agent's personality cues in dialogue match the attributes as expected. These

| Paper | Eval. Granularity | Reference | Objective Eval? | Effect Size? | Multi Turn? |
|-------|-------------------|-----------|-----------------|--------------|-------------|
| Argyle et al. (2023) | Population-Level | Human surveys | ✓ | ✗ | ✗ |
| Wang et al. (2023b) | Individual-Level | Self-elicited beliefs | ✗ | ✗ | ✗ |
| Huang et al. (2024) | Individual-Level | Social norms | ✗ | ✗ | ✗ |
| Bhandari et al. (2025) | Individual-Level | Assigned persona | ✗ | ✗ | ✗ |
| Hou et al. (2025) | Both | Expert policy | ✓ | ✗ | ✓ |
| Our Work | Both | Self-elicited beliefs | ✓ | ✓ | ✓ |

Table 1: Comparison of evaluation frameworks for LLM behavioral consistency in simulations of human behavior. **Evaluation Regime**: Whether consistency is assessed at the population level, individual agent level, or both. **Reference**: The ground truth standard against which LLM behavior is compared (e.g., human survey data, theoretical models, agent's own stated beliefs). **Objective Eval?**: Whether evaluation uses quantitative metrics (Objective) versus LLM-based judgment. **Effect Size?**: Whether the framework quantifies the magnitude of behavioral relationships, not just their direction. **Multi Turn?**: Whether simulations involve extended interactions across multiple rounds.

evaluation methods act as valuable benchmarks, but they share two key limitations: they rely heavily on external reference data, which can be limited or difficult to obtain, and they evaluate agent behavior only after simulation, making them fundamentally post-hoc.

In addition to evaluating role-playing agents individually, frameworks such as VACSIM (Hou et al., 2025) evaluate emergent population-level phenomena, like opinion dynamics in social networks. Such ambitious simulations highlight the potential impact of LLM-based role-play can have in influencing real-life policy decisions, but also raise the stakes: undetected errors can propagate, leading to misleading or harmful conclusions. Here, too, evaluation is typically performed after data is generated, reinforcing the post-hoc nature of current practices.

As summarized in Table 1, nearly all existing evaluation methods for role-playing agents are post-hoc. This is problematic because, as Orgad et al. (2024) show, LLMs may encode accurate knowledge internally but often apply it inconsistently across different contexts. Since validation only takes place after simulations are complete, errors in synthetic data can go undetected when outcomes cannot be independently verified. This reactive approach not only increases costs but also allows flawed data to influence downstream analyses.

## 3 Experimental Framework

To rigorously evaluate belief-behavior consistency, we implement our framework using the Trust Game as a testbed (Berg et al., 1995). In this game, a role-playing LLM agent (the Trustor) decides how much money to send to another player (the Trustee). The amount sent is tripled, and the Trustee then decides how much to return. We simulate this scenario using multiple LLMs (Llama 3.1 8B/70B, (Grattafiori et al., 2024), Gemma 2 27B, (Team et al., 2024)) and a diverse set of synthetic personas with distinct demographic and personality attributes (see Section 3.2).

We structure our experiments at two complementary analysis levels. At the *population level*, we measure consistency by comparing the elicited marginal correlations between persona attributes and simulated behaviors (e.g., how age affects decisions in the Trust Game). At this level, we also test the effectiveness of using a simple, prompt-based approach to *impose* a desired belief onto the role-playing agent. At the *individual level*, we test if the LLM can accurately forecast its own actions when role-playing an individual persona across multiple rounds against clearly defined Trustee strategies.

### 3.1 Trust Game Environment

The canonical two-player *Trust Game* proceeds as follows. At the start of each round, Player A (also known as the *Trustor*) receives an endowment $E$ ($10, unless otherwise noted). The Trustor chooses an amount $s \in [0, E]$ to send to Player B (the *Trustee*). The amount sent is tripled before reaching the Trustee, who then decides how much $r \in [0, 3s]$ to return to the Trustor, keeping the remainder.

We simulate the Trustor using a role-playing LLM agent, providing it with a specific persona and the rules of the game. The Trustor agent is the first to act in the Trust Game; thus, its decision depends solely on its assigned persona, not on any prior moves by the Trustee. This design eliminates confounding effects from the other player's behavior. In single-turn scenarios (Section 4), we measure only how much money the Trustor chooses to send. In multi-turn scenarios (Section 5), the Trustee's responses are fully determined by fixed archetypes, as described in the individual-level experiment section (Section 5).

The role-playing prompt explicitly states the Trustor's role, the available action space (integer dollar amounts), and asks for a numeric response. We sample the model's output distribution with temperature scaling. Unless otherwise specified, we use $T=0.05$ for replicability. Details of our output extraction and parsing methodology are provided in Appendix E. We additionally vary the endowment $E$ to test robustness across initial amounts (prompt template provided in Appendix D.1).

### 3.2 Participant Generation

We construct a synthetic participant pool by augmenting the existing GENAGENTS persona bank (Park et al., 2024). In addition to the demographic and social attributes present in GENAGENTS (including age, education, ethnicity, gender, income, political ideology, geographic region), we augment them with independently sampled values for the Big Five personality dimensions—openness, conscientiousness, extraversion, agreeableness, and neuroticism—drawn at random (DeYoung et al., 2007). We include these five personality dimensions because psychological research demonstrates their robust associations with interpersonal trust (Sharan & Romano, 2020; Bartosik et al., 2021; Evans & Revelle, 2008). The purpose of the augmentation is to increase the variability of attribute-behavior effect sizes in the Trust Game. We represent each persona as an attribute vector, $p_i = (t_1^{(i)}, t_2^{(i)}, \ldots, t_K^{(i)})$, where $t_k^{(i)}$ denotes the value of the $k$-th attribute for persona $i$. Attributes are a mix of categorical and ordinal types. All experimental results in this paper use our test split of the GENAGENTS persona bank; the training and validation splits were used only for prompt template design and pipeline tuning. Appendix A provides a complete list of attribute types and details of the dataset splits.

### 3.3 Organization of Experiments

Our experimental study proceeds in two parts. First, Section 4 considers: (a) the evaluation of population-level belief-behavior consistency, assessing whether statistical patterns in simulated behavior align with elicited beliefs; and (b) whether consistency improves when LLMs are *self-conditioned* (receiving their own elicited beliefs as additional context in the role-playing prompt), or deteriorates under the use of *imposed priors* (when agents are given modified priors that systematically diverge from their original beliefs to test controllability). Second, Section 5 presents an assessment of individual-level belief-behavior consistency, comparing an agent's forecasted and enacted actions across multiple rounds of the Trust Game against fixed Trustee archetypes. Together, these analyses offer a multi-scale view of belief-behavior consistency in LLM-based simulations.

## 4 Population-Level Consistency

Population-level belief-behavior consistency measures how closely a model's elicited beliefs about belief-behavior relationships match the patterns observed in simulation. We assess this consistency by comparing the model's elicited predictions with the correlations between persona attributes and simulated trust behavior, marginalized over all other persona attributes except the one whose belief is under consideration. Our experimental setup systematically evaluates how key design choices in the belief elicitation process affect the consistency between elicited beliefs and subsequent behaviors. For each belief elicitation strategy, we use N = 50 (Wang et al. (2023b), who have a similar experimental setup as ours, use N = 32) personas to compute our results.

| Strategy (S) | Elicitation Target | Context? | Description |
|---|---|---|---|
| NoCtx+Tr | Interpersonal Trust | ✗ | Rank trait levels by their anticipated impact on "Interpersonal Trust," without mentioning the Trust Game. |
| Ctx+Tr | Interpersonal Trust | ✓ | Same ranking question, but preceded by the full Trust Game instructions and role description. |
| Ctx+$ | Dollars sent | ✓ | Estimate, for each trait level, the mean and standard deviation of dollars sent in the Trust Game. |

Table 2: Population-level belief elicitation strategies. "Elicitation Target" specifies whether the model ranks trait effects against *interpersonal trust* or against the amount of money sent by the Trustor; "Context?" denotes whether full game instructions were provided during belief elicitation.

## 4.1 Belief Elicitation and Evaluation Metrics

We query the language model to determine how it believes a persona attribute of interest (for example, participant age) influences decisions in the Trust Game. We apply three elicitation strategies (see Table 2). For each elicitation strategy $S$, we derive two structured outputs:

- **Belief ranking** $\pi_t^{(S)}$: an ordered list of values of attribute $t$, ranked by their predicted effect on the average amount of money transferred by the Trustor, in descending order.

- **Belief effect size** ($\widehat{\eta}_t^{2(S)}$): an estimate of the proportion of variance in the Trustor's decisions attributable to differences among the $K$ groups defined by attribute $t$. We compute $\widehat{\eta}_t^{2(S)}$ using the eta-squared ($\eta^2$) effect size from analysis of variance (ANOVA; Girden, 1992), where the groups correspond to attribute levels.

**Behavioral outcomes:** We simulate the Trust Game for each persona $p_i$, obtaining the transfer amount $\{s_i\}$. The prompt template used for role-playing is given in Appendix B.1. From these we derive:

- **Behavioral ranking** $\pi_t^*$: an ordered list of the discrete values of attribute $t$, ranked by the observed mean $ transfer among personas possessing each value, in descending order.

- **Behavioral effect size** $\eta_t^2$: the proportion of variance in the simulated transfers attributable to differences among the levels of attribute $t$, computed via eta-squared from a one-way ANOVA grouping by those levels.

**Belief-Behavior Consistency Metrics:** Finally, we assess belief-behavior consistency via two complementary statistics. First, we compute the Spearman correlation, which captures how well the elicited ranking matches the behavioral ordering. Second, we measure the absolute effect-size discrepancy quantifying the difference between predicted and observed effect sizes.

$$\underbrace{\rho_t^{(S)} = \text{Spearman}\left(\pi_t^{(S)}, \pi_t^*\right)}_{\substack{\text{ranking} \\ \text{consistency}}} \quad \text{and} \quad \underbrace{\Delta\eta_t^{2(S)} = \left|\widehat{\eta}_t^{2(S)} - \eta_t^2\right|}_{\substack{\text{effect-size} \\ \text{consistency}}}.$$

## 4.2 Population-Level Consistency Analysis

Table 3 reports the median Spearman correlation and median absolute effect size difference for each LLM and elicitation strategy. Using the median for both metrics reduces sensitivity to outlier attributes.

**Providing task-specific context during belief elicitation does not enhance belief-behavior consistency.** We compare two belief elicitation strategies that differ only in the presence of explicit Trust Game instructions when asking the model to predict how persona attributes influence transfer amounts: one strategy provides full game rules and the agent's role,

| Strategy | Attribute | G22 | | L7I | | L8I | |
|---|---|---|---|---|---|---|---|
| | | $\|\Delta\eta^2\|$ | $\rho$ | $\|\Delta\eta^2\|$ | $\rho$ | $\|\Delta\eta^2\|$ | $\rho$ |
| NoCtx+Tr | Age | 0.02 | -0.20 | 0.00 | -1.00 | 0.14 | 0.00 |
| | Conscientiousness | 0.01 | 1.00 | 0.07 | 0.50 | 0.05 | 1.00 |
| | Family Structure | 0.07 | 0.83 | 0.00 | 0.49 | 0.09 | -0.26 |
| | Highest Degree | 0.02 | -0.10 | 0.02 | -0.60 | 0.09 | -0.30 |
| | Openness | 0.06 | 1.00 | 0.02 | 1.00 | 0.06 | 1.00 |
| | Political Views | 0.01 | 0.40 | 0.01 | 0.40 | 0.09 | 1.00 |
| | Same Residence | 0.02 | 0.50 | 0.00 | 0.50 | 0.15 | -0.50 |
| | US Citizen | 0.01 | 1.00 | 0.01 | 1.00 | 0.02 | 1.00 |
| | Work Status | 0.03 | 0.80 | 0.00 | -1.00 | 0.02 | 0.80 |
| | **Median** | **0.02** | **0.80** | **0.01** | **0.49** | **0.09** | **0.80** |
| Ctx+Tr | Age | 0.09 | -0.40 | 0.00 | 0.80 | 0.02 | 0.40 |
| | Conscientiousness | 0.01 | 1.00 | 0.07 | 0.50 | 0.11 | 1.00 |
| | Family Structure | 0.07 | -0.37 | 0.07 | 0.49 | 0.09 | -0.26 |
| | Highest Degree | 0.04 | -0.10 | 0.02 | -0.60 | 0.03 | -0.60 |
| | Openness | 0.06 | 1.00 | 0.02 | 1.00 | 0.13 | 1.00 |
| | Political Views | 0.08 | 0.40 | 0.01 | 0.40 | 0.16 | 1.00 |
| | Same Residence | 0.08 | 0.50 | 0.00 | 0.50 | 0.09 | -0.50 |
| | US Citizen | 0.01 | 1.00 | 0.01 | 1.00 | 0.02 | 1.00 |
| | Work Status | 0.03 | 0.40 | 0.00 | -1.00 | 0.02 | 0.40 |
| | **Median** | **0.06** | **0.40** | **0.01** | **0.50** | **0.09** | **0.40** |
| Ctx+$ | Age | 0.06 | 0.80 | 0.03 | 1.00 | 0.21 | 0.40 |
| | Conscientiousness | 0.40 | 1.00 | 0.49 | 0.50 | 0.46 | 1.00 |
| | Family Structure | 0.01 | 0.83 | 0.03 | 0.37 | 0.14 | -0.26 |
| | Highest Degree | 0.13 | -0.10 | 0.16 | -0.60 | 0.30 | -0.30 |
| | Openness | 0.55 | 1.00 | 0.54 | 1.00 | 0.56 | 1.00 |
| | Political Views | 0.15 | 1.00 | 0.21 | 1.00 | 0.05 | 1.00 |
| | Same Residence | 0.05 | 0.50 | 0.02 | 0.50 | 0.20 | -0.50 |
| | US Citizen | 0.01 | 1.00 | 0.00 | 1.00 | 0.23 | -1.00 |
| | Work Status | 0.10 | 0.80 | 0.03 | -0.40 | 0.13 | 0.00 |
| | **Median** | **0.10** | **0.83** | **0.03** | **0.50** | **0.21** | **0.00** |

Table 3: Population-Level Self-Consistency Analysis: Effect Size Difference ($|\Delta\eta^2|$) and Spearman Correlation ($\rho$) Across Models and Strategies. Lower $|\Delta\eta^2|$ indicates better portability (smaller effect size differences). Higher $\rho$ indicates better correlation preservation across contexts.

while the other omits all task-specific context (see Appendix C.1.2 and Appendix C.1.1 respectively for prompt templates). Across all LLMs and attributes, supplying Trust Game context during belief elicitation failed to increase Spearman rank correlation or reduce the absolute effect-size discrepancy compared to context-free elicitation. These findings contradict prior evidence that contextual prompts reliably shift LLM outputs away from pretraining biases (Tao et al., 2024), indicating that providing additional context alone does not improve the agreement between elicited attribute-behavior beliefs and the agent's simulated transfer decisions.

**Elicitation target affects different aspects of belief-behavior consistency.** We compare belief elicitation strategies that target different constructs: behavioral outcomes directly (Ctx+$, asking for dollar amounts sent) versus latent psychological constructs (Ctx+Tr, NoCtx+Tr, asking about interpersonal trust levels). These strategies exhibit complementary strengths across our two consistency metrics. For *rank ordering consistency* (Spearman correlation), the behavioral outcome strategy (Ctx+$) yields more accurate attribute rankings than construct-focused strategies, as it directly mirrors the simulation task. However, for *effect size consistency*, construct-focused strategies (Ctx+Tr, NoCtx+Tr) produce estimates more closely aligned with behavioral values, while the behavioral outcome strategy (Ctx+$) systematically overestimates effect magnitudes. This suggests that LLMs encode attribute-behavior relationships differently when queried about psychological abstractions versus concrete behavioral predictions.

## 4.3 Conditioning Experiments: Can Priors Be Reinforced or Overriden?

Imposing researcher-specified priors is essential for correcting biases (e.g., adjusting for an overrepresented demographic group in training data to prevent skewed outputs; Wang et al.,

2025), testing theoretical predictions in specialized domains (e.g., assuming introverts speak more words per minute than introverts; Argyle et al., 2023), and exploring counterfactuals that diverge from elicited beliefs (e.g., simulating an alternate cultural context where societal norms differ; Zhang et al., 2025). If it turns out that the ability to impose an arbitrary belief is limited, the utility of LLMs as flexible tools for hypothesis testing is constrained. We therefore investigate whether LLM behavior can be controlled via belief conditioning: can reinforcing an agent's own beliefs improve consistency, and can external priors override the model's defaults?

We compare three conditions: (1) a baseline case with no conditioning (prompt template in Appendix C.1.2), (2) a *self-conditioned* case where agents receive their own elicited beliefs as context (prompt template in Appendix C.2.3), and (3) two *imposed priors* cases where we supply priors that systematically diverge from the model's elicited beliefs (also using the prompt template in Appendix C.2.3, but with a perturbation procedure applied to the elicited beliefs). For each approach, we evaluate both ranking consistency (using Spearman correlation) and effect size consistency (using mean absolute error, Mean Absolute Error (MAE)).

| Belief Conditioning | Gemma 2 27B | Llama 3.1 70B | Llama 3.1 8B |
|---|---|---|---|
| Unconditioned | 0.40 | 0.50 | 0.40 |
| Self-Conditioned | 0.00 | 0.80 | 1.00 |
| Weak Perturbation ($\rho = 0.80$) | 0.08 | 0.30 | -0.14 |
| Strong Perturbation ($\rho = 0.20$) | 0.14 | 0.20 | 0.40 |

Table 4: Belief conditioning effectiveness across models and methods. Values are Spearman correlations ($\rho$; higher is better) between the imposed prior and simulated behavior. **Unconditioned** reports baseline consistency. **Self-conditioned** supplies each model's own elicited beliefs as context. **Weak** and **strong perturbation** conditions apply modified priors constructed to have $\rho = 0.80$ or $\rho = 0.20$ correlation, respectively, with the original elicited beliefs. Self-conditioning shows highly model-dependent effects; even weakly perturbed priors can substantially disrupt belief–behavior alignment.

### 4.3.1 Results

**Self-conditioning enhances consistency in Llama models, but not in Gemma 2 27B.** Self-conditioning increases Spearman's $\rho$ for Llama 3.1 70B from 0.50 to 0.80, and for Llama 3.1 8B from 0.40 to 1.00, but Gemma 2 27B drops from 0.40 to less than 0.01 (Table 4). In contrast, Gemma 2 27B's $\rho$ drops from 0.40 to less than 0.01 under the same procedure. These results indicate that self-conditioning effectively aligns Llama models' behavior with their elicited beliefs, whereas Gemma 2 27B remains unresponsive to in-context belief prompts, showing mixed results across architectures.

**Imposed priors systematically undermine consistency.** Table 4 shows that imposing modified priors—created by perturbing each model's original elicited beliefs—sharply reduces rank ordering consistency across all models. When the imposed priors are only weakly perturbed from the elicited beliefs (constructed to have $\rho = 0.80$ with the original), the Spearman correlation for Llama 3.1 70B drops from 0.50 to 0.30, for Llama 3.1 8B from 0.40 to $-0.14$, and for Gemma 2 27B from 0.40 to 0.08. When the imposed priors are strongly perturbed (constructed to have $\rho = 0.20$ with the original), consistency declines further for Llama 3.1 70B (dropping from 0.50 to 0.20), while Llama 3.1 8B returns to its unconditioned baseline (0.40), and Gemma 2 27B increases slightly (from 0.08 under weak perturbation to 0.14). These results indicate that even modest divergence between imposed and elicited priors can substantially impair belief-behavior alignment, with the magnitude and direction of the effect depending on model architecture.

## 5 Individual-Level Consistency

Our second investigation tests the LLM's ability to forecast its own future behavior while role-playing as a specific individual. This *individual-level* analysis contrasts with the ear-

lier *population-level* analysis, which examined correlations between persona attributes and behaviors across many personas (see Appendix C.3.1 for the belief elicitation prompt and Appendix C.4 for sample beliefs). Here, we focus on how role-play unfolds over time, allowing us to study how forecasting accuracy degrades with longer horizons. In contrast, the population-level analysis is limited to single-round simulations to avoid confounding from conditioning on a specific Trustee archetype. For individual-level role-playing, we use the ReAct framework (Yao et al., 2023), which interleaves reasoning and action steps to structure multi-step decision-making (prompt in Appendix B.2).

## 5.1 Trustee Archetypes For Belief Elicitation

In the Trust Game, the only source of stochasticity is the Trustee's behavior. By fixing the Trustee's strategy via a well-defined archetype, we fully specify the simulation environment. Following agent-based modeling literature (Chopra et al., 2024), we define simple, interpretable archetypes of the opponent player (Trustee) to ensure straightforward forecast evaluation, while avoiding the combinatorial explosion of conditioning on every possible response of the opponent.

For example, querying an LLM about its predicted action in round five of a Trust Game against a consistently uncooperative opponent requires us to pass in a precise, standardized description of that opponent's behavior to the LLM's prompt. Without such structure, comparisons between forecasted beliefs and simulated behavior would be unfair, as the LLM would be making predictions without access to the contextual information necessary to ground its responses.

We define three Trustee archetypes: $\mathcal{M}_1$, $\mathcal{M}_3$, and $\mathcal{M}_5$. Each archetype corresponds to a Trustee that returns $\$i$, where $i$ is the subscript in the archetype's name (i.e., $\mathcal{M}_1$ returns $\$1$, $\mathcal{M}_3$ returns $\$3$, and $\mathcal{M}_5$ returns $\$5$), or their total available funds if less (see Appendix C.3 for details). These archetypes provide simple, controlled baselines for forecast elicitation and evaluation.

## 5.2 Individual-Level Self-Consistency

We evaluate a single persona over $R$ rounds of the Trust Game (for $r = 1, \ldots, R$) against a fixed trustee archetype. In each round $r$, the model first forecasts its send amount $\hat{s}_r$, and then we simulate that round to observe the actual send $s_r$. Forecasting accuracy across all rounds is measured by the mean absolute error. This MAE serves as our individual-level self-consistency metric:

$$\text{MAE} = \frac{1}{R} \sum_{r=1}^{R} |\hat{s}_r - s_r|.$$

At the individual level, we assess an LLM's ability to forecast its own actions from crafted summary state descriptions (see Appendix C.3). We prompt the model to predict its behavior as Trustor in the Trust Game and measure accuracy via mean absolute error (MAE) between predicted and actual transfer amounts. This differs from our population-level evaluation which uses Spearman correlations for attribute rankings and eta-squared for effect size measures (as detailed in Section 4).

## 5.3 Results

**Forecasting accuracy degrades over longer horizons.** We observe a near-monotonic increase in MAE as the forecast horizon extends from one to six rounds, indicating that the belief-behavior consistency reduces when predicting actions $\hat{s}_r$ further removed from the current game state (Fig. 1). This pattern holds across most model–archetype pairs, with the exception of Llama 3.1 8B under the $\mathcal{M}_3$ and $\mathcal{M}_5$ archetypes—where MAE remains flat or slightly decreases—suggesting that longer horizons could introduce accumulating uncertainty, which undermines a model's forecasting ability.

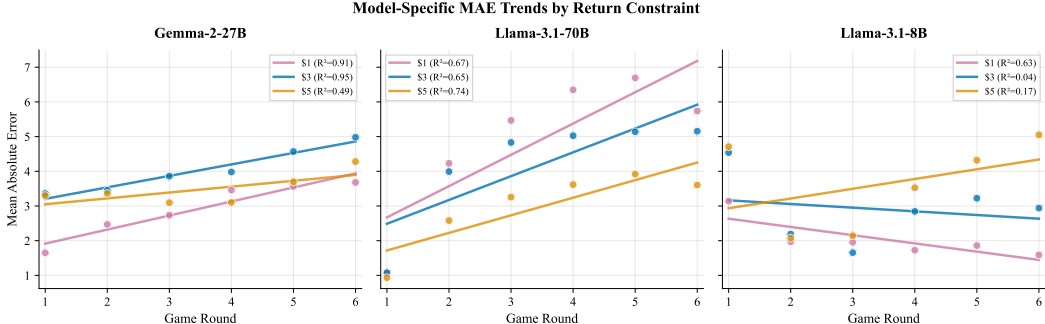

Figure 1: Trust Game: Model consistency across rounds, stratified by return constraint ($1, $3, $5). Each regression line represents MAE over six repeated rounds of the Trust Game for a given LLM-archetype combination. Lower MAE indicates higher forecasting consistency.

## 6   Limitations, Discussion, and Implications

**Reasoning models may improve belief-behavior consistency.**   The systematic inconsistencies we observe may stem from the rapid, single-pass inference typical of traditional LLMs. Recent reasoning models of the likes of DeepSeek-R1 (Guo et al., 2025) and OpenAI o1 and o3 (Jaech et al., 2024) employ extended reasoning processes that could potentially bridge the gap between belief elicitation and behavioral simulation. These models' ability to engage in multi-step reasoning and self-reflection during inference may enable more coherent application of stated beliefs to subsequent actions.

**Limits of in-context conditioning for controllability.**   While self-conditioning improves consistency in some Llama models, imposed priors tend to undermine it across architectures. This suggests a potential limitation: in-context prompting may struggle to override entrenched model priors, which could limit researchers' ability to test alternative theories or correct biases. Future work might explore knowledge editing (Wang et al., 2023a; Orgad et al., 2024) or inference-time steering (Li et al., 2023; Lamb et al., 2024; Minder et al., 2025) for more robust belief control.

**Generalization beyond the Trust Game.**   While our evaluation framework and belief-behavior consistency measures are general and domain-agnostic, we demonstrated their utility through the Trust Game, which provides a simple, well-structured environment that served as an ideal testbed for this initial study. However, real-world simulations often involve richer social contexts and more nuanced agent-goals. In future work, we aim to apply our framework to multi-agent environments, open-ended dialogues, or temporally extended tasks to evaluate how these inconsistencies manifest in more complex settings.

## 7   Conclusion

We investigate belief-behavior consistency in LLM-based role-playing agents using the Trust Game, revealing systematic inconsistencies between models' stated beliefs and simulated behaviors at both population and individual levels. Our evaluation framework identifies these issues before costly deployment by eliciting beliefs as a diagnostic tool. Key findings show that providing task context during belief elicitation does not improve consistency, self-conditioning helps some models while imposed priors generally undermine alignment, and forecasting accuracy degrades over longer horizons. These results highlight fundamental limitations in current LLM role-playing approaches and emphasize the need for robust internal consistency evaluation before using these systems as scientific instruments.

## Ethics Statement

This work uses publicly available large language models (Llama 3.1, Gemma 2) and synthetic personas based on the GENAGENTS dataset, with no human subjects or private data involved. All simulations and elicited beliefs were generated using artificial agents.

We highlight limitations in belief-behavior consistency of LLM-based role-playing agents and propose methods to evaluate internal consistency prior to deployment. While some experiments include intentionally imposed priors to test model controllability, these do not reflect normative claims about group differences.

No sensitive or high-risk applications are addressed. The findings are intended to improve the reliability and transparency of synthetic simulations and are not substitutes for real-world behavioral data. Code and prompts will be released to support reproducibility and responsible use.

## Acknowledgements

This work is supported in part by the National Science Foundation and the Institute of Education Sciences, U.S. Department of Education, through Award #2229612 (National AI Institute for Inclusive Intelligent Technologies for Education). Any opinions, findings, and conclusions or recommendations expressed in this material are those of the author(s) and do not necessarily reflect the views of National Science Foundation or the U.S. Department of Education.

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

## A  Persona Attributes

Table 5 summarizes the full set of demographic, personality, and contextual attributes that define each synthetic persona used throughout our experiments.

## B  Role-Playing Prompts

### B.1  Population-Level Role-Playing

```
===== TRUST GAME INSTRUCTIONS =====

YOU ARE PLAYING A TRUST GAME AS THE FIRST PLAYER (TRUSTOR). READ THESE
INSTRUCTIONS CAREFULLY:

1. YOU HAVE EXACTLY $$${initial_amount} TO USE IN THIS GAME. NO MORE, NO
     LESS.
2. YOU MUST DECIDE HOW MUCH OF YOUR $$${initial_amount} TO TRANSFER TO
     ANOTHER PLAYER.
3. ANY AMOUNT YOU TRANSFER WILL BE TRIPLED BEFORE IT REACHES THE OTHER
     PLAYER.
4. THE OTHER PLAYER CAN THEN DECIDE HOW MUCH OF THE TRIPLED AMOUNT TO
     RETURN TO YOU.
5. YOU CANNOT TRANSFER MORE THAN $$${initial_amount} UNDER ANY
     CIRCUMSTANCES.

===== YOUR CHARACTER PROFILE =====

$persona_details

$theoretical_props

===== FINAL REMINDERS =====

1. YOU HAVE EXACTLY $$${initial_amount} AVAILABLE TO TRANSFER.
2. YOUR TRANSFER WILL BE TRIPLED.
3. THE OTHER PLAYER DECIDES HOW MUCH TO RETURN TO YOU.
4. STATE YOUR FINAL DECISION CLEARLY AS A DOLLAR AMOUNT BETWEEN $$0 AND
     $$${initial_amount}.
5. Condition your action on your persona profile.

How much of your $$${initial_amount} will you transfer?

Your answer needs to include your BELIEF, DESIRE and INTENTION.
End with 'Finally, I will give ___ dollars'.
```

### B.2  Individual-Level Role-Playing Prompt

The following is the prompt template used for multi-round Trustor role-play in the Trust Game using the ReAct framework. This template is used to elicit the Trustor's decision and reasoning at each round, incorporating persona, game state, and history.

```
You are a Trustor in the Trust Game with the following persona:

${persona}

===== TRUST GAME INSTRUCTIONS =====

YOU ARE PLAYING A MULTI-ROUND TRUST GAME AS THE FIRST PLAYER (TRUSTOR).
     READ THESE INSTRUCTIONS CAREFULLY:

1. THIS IS ROUND $round_number OF $total_rounds.
2. YOU HAVE EXACTLY $$${remaining_amount} TO USE IN THIS ROUND. NO MORE,
```

| *Trait* | *Possible Values* |
|---|---|
| **Age**[te] | 18-29[te], 30-44[te], 45-64[te], 65+[te] |
| **Agreeableness**[tr] | High[tr], Low[tr], Medium[tr] |
| **Communication Quality**[v] | Excellent[v], Neutral[v], Poor[v] |
| **Conscientiousness**[te] | High[te], Low[te], Moderate[te] |
| **Cultural Norms**[tr] | Collectivist[tr], Hybrid[tr], Individualist[tr] |
| **Ethnicity**[v] | Arab[v], Black African[v], East Asian[v], Indigenous American[v], Latino[v], Other[v], South Asian[v], Southeast Asian[v], White[v] |
| **Extraversion**[v] | Ambivert[v], Extraverted[v], Introverted[v] |
| **Family Structure At 16**[tr,te] | Armed forces[tr], Both parents[te], Divorce[tr], Foster care[te], Grandparents[te], Institution[tr], Lived with parents[tr], Other guardian[te], Single parent - father[te], Single parent - mother[te] |
| **General Activity Level**[v] | Lightly active[v], Moderately active[v], Sedentary[v], Unstructured[v], Very active[v] |
| **Highest Degree Received**[v,tr,te] | Associate/junior college[tr,te], Bachelor's[tr,te], Bachelor's degree[v], Graduate[tr,te], Graduate or professional degree[v], High school[tr,te], High school diploma or GED[v], Less than high school[tr,te], No high school diploma[v], Some college or associate degree[v] |
| **Marital Status**[tr] | Divorced[tr], Married[tr], Never married[tr], Separated[tr], Widowed[tr] |
| **Neuroticism**[v] | High[v], Low[v], Moderate[v] |
| **Openness To Experience**[te] | High[te], Low[te], Moderate[te] |
| **Political Views**[tr,te] | Conservative[tr], Extremely conservative[tr,te], Extremely liberal[tr,te], Liberal[tr], Moderate, middle of the road[tr], Slightly conservative[te], Slightly liberal[te] |
| **Race**[tr] | Black[tr], Other[tr], White[tr] |
| **Religion**[tr,te] | Buddhism[tr], Christian[tr], Hinduism[tr], Jewish[te], Muslim/Islam[te], None[te], Orthodox-Christian[te], Protestant[te] |
| **Same Residence Since 16**[te] | Different state[te], Same city[te], Same state, different city[te] |
| **Sex**[tr] | Female[tr], Male[tr] |
| **Total Wealth**[tr] | Less than \$20,000[tr], \$20,000–\$75,000[tr], \$75,000–\$250,000[tr], \$250,000–\$1 million[tr], \$1 million–\$5 million[tr], Above \$5 million[tr] |
| **Type Of Disability If Any**[v] | Cognitive or learning disability[v], Mental health condition[v], None[v], Physical disability[v], Sensory disability (e.g., vision or hearing)[v] |
| **Us Citizenship Status**[te] | A U.S. citizen[te], Not a U.S. citizen[te] |
| **Work Status**[tr,te] | Full time[tr], In school[te], Keeping house[te], Other[te], Part time[tr], Retired[te], Temporarily not working[tr], Unemployed[tr] |

Table 5: Traits and possible values used in our synthetic persona dataset, which extends the GENA-GENTS dataset with additional demographic, personality, and trust-related traits. The superscripts indicate which data splits (tr=train, v=val, te=test) each trait and value appears in.

```
      NO LESS.
3. YOU MUST DECIDE HOW MUCH OF YOUR $$${remaining_amount} TO TRANSFER
   TO THE OTHER PLAYER.
4. ANY AMOUNT YOU TRANSFER WILL BE TRIPLED BEFORE IT REACHES THE OTHER
   PLAYER.
5. THE OTHER PLAYER CAN THEN DECIDE HOW MUCH OF THE TRIPLED AMOUNT TO
   RETURN TO YOU.
6. YOU CANNOT TRANSFER MORE THAN $$${remaining_amount} UNDER ANY
   CIRCUMSTANCES.

===== GAME HISTORY =====

Previous rounds:
$round_history

$theoretical_props

How much of your $$${remaining_amount} will you transfer in this round?
Your answer needs to include your THOUGHT, ACTION, and OBSERVATION.

${format_instructions}
```

This template ensures that the agent is provided with all relevant context, including persona, round number, available funds, game history, and any theoretical properties, and is instructed to respond with a structured reasoning process (THOUGHT, ACTION, OBSERVATION).

## C   Belief Elicitation Prompts and Sample Outputs

### C.1   Population-Level Belief Elicitation

This appendix provides example prompts used for belief elicitation, illustrating both context-free and context-specific (contextful) approaches.

### C.1.1   NOCTX+TR *Belief Elicitation Prompt*

```
You are a researcher studying interpersonal trust.

For the trait "general_activity_level", with the following possible
values:
1. Lightly active
2. Moderately active
3. Sedentary
4. Unstructured
5. Very active

Please rank these values in order from highest to lowest in terms of
their association with interpersonal trust, using the format "Value1 >
Value2 > Value3" where ">" means "has higher interpersonal trust than".

Your response should include an ordering of all the values listed above
and a brief explanation of your reasoning.

The output should be formatted as a JSON instance that conforms to
the JSON schema below.

As an example, for the schema {"properties": {"foo": {"title": "Foo",
"description": "a list of strings", "type": "array", "items":
{"type": "string"}}}, "required": ["foo"]}
the object {"foo": ["bar", "baz"]} is a well-formatted instance of the
schema. The object {"properties": {"foo": ["bar", "baz"]}} is not
well-formatted.
```

```
Here is the output schema:
...
```

### C.1.2  CTX+TR *Belief Elicitation Prompt*

```
You are a researcher studying interpersonal trust.

For the attribute "general_activity_level", with the following possible
values:
1. Lightly active
2. Moderately active
3. Sedentary
4. Unstructured
5. Very active

Please rank these values in order from highest to lowest in terms of
their association with interpersonal trust, using the format "Value1 >
Value2 > Value3" where ">" means "has higher interpersonal trust than".

Your response should include an ordering of all the values listed above
and a brief explanation of your reasoning.

Consider this simulation context while answering: YOU ARE PLAYING A TRUST
GAME AS THE FIRST PLAYER (TRUSTOR). READ THESE INSTRUCTIONS CAREFULLY:

1. YOU HAVE BEEN GIVEN $10 AS YOUR INITIAL AMOUNT OF MONEY TO USE IN
   THIS GAME. NO MORE, NO LESS.
2. YOU MUST DECIDE HOW MUCH OF YOUR INITIAL AMOUNT OF MONEY TO TRANSFER
   TO ANOTHER PLAYER. YOU CAN SEND ANY AMOUNT FROM $0 TO $10.
3. ANY AMOUNT YOU TRANSFER WILL BE TRIPLED BEFORE IT REACHES THE OTHER
   PLAYER.
4. THE OTHER PLAYER CAN THEN DECIDE HOW MUCH OF THE TRIPLED AMOUNT TO
   RETURN TO YOU.
5. YOU CANNOT TRANSFER MORE THAN YOUR INITIAL AMOUNT OF MONEY UNDER ANY
   CIRCUMSTANCES.

The output should be formatted as a JSON instance that conforms to the
JSON schema below.

As an example, for the schema {"properties": {"foo": {"title": "Foo",
"description": "a list of strings", "type": "array", "items":
{"type": "string"}}}, "required": ["foo"]}
the object {"foo": ["bar", "baz"]} is a well-formatted instance of the
schema. The object {"properties": {"foo": ["bar", "baz"]}} is not
well-formatted.

Here is the output schema:
...
```

### C.1.3  CTX+$ *Belief Elicitation Prompt*

```
Imagine an experiment where we control for various factors and variables
while examining the effect of general_activity_level. Your task is to
simulate the outcomes of this experiment.

Consider this simulation context while answering: YOU ARE PLAYING A TRUST
GAME AS THE FIRST PLAYER (TRUSTOR). READ THESE INSTRUCTIONS CAREFULLY:

1. YOU HAVE BEEN GIVEN $10 AS YOUR INITIAL AMOUNT OF MONEY TO USE IN
   THIS GAME. NO MORE, NO LESS.
2. YOU MUST DECIDE HOW MUCH OF YOUR INITIAL AMOUNT OF MONEY TO TRANSFER
```

```
         TO ANOTHER PLAYER. YOU CAN SEND ANY AMOUNT FROM $0 TO $10.
3. ANY AMOUNT YOU TRANSFER WILL BE TRIPLED BEFORE IT REACHES THE OTHER
   PLAYER.
4. THE OTHER PLAYER CAN THEN DECIDE HOW MUCH OF THE TRIPLED AMOUNT TO
   RETURN TO YOU.
5. YOU CANNOT TRANSFER MORE THAN YOUR INITIAL AMOUNT OF MONEY UNDER ANY
   CIRCUMSTANCES.

For each level of general_activity_level (Lightly active, Moderately
active, Sedentary, Unstructured, Very active) listed below, please
provide your best estimate of:

- The average (mean) dollar amount these individuals would send in a
  trust game.
- The typical variability in the amount sent (as a standard deviation).

Assume each group consists of 100 individuals to help you better estimate
both the mean and standard deviation.

The output should be formatted as a JSON instance that conforms to the
JSON schema below.

As an example, for the schema {"properties": {"foo": {"title": "Foo",
"description": "a list of strings", "type": "array", "items":
{"type": "string"}}}, "required": ["foo"]}
the object {"foo": ["bar", "baz"]} is a well-formatted instance of the
schema. The object {"properties": {"foo": ["bar", "baz"]}} is not
well-formatted.

Here is the output schema:
...
```

## C.2   Sample Population-Level Beliefs

### C.2.1   NOCTX+TR and CTX+TR

```
{
  "political_views": {
    "ranking_descending": [
      "Extremely liberal",
      "Slightly liberal",
      "Slightly conservative",
      "Extremely conservative"
    ],
    "omnibus_effect_size": "medium",
    "contrast_effect_size": "small",
    "ordering_explanation": "Individuals with liberal political views tend
        to be more open-minded and accepting of diversity, which can
        foster a sense of trust in interpersonal relationships. In
        contrast, individuals with conservative political views may be
        more skeptical of change and less accepting of diversity, leading
        to lower levels of interpersonal trust.",
    "omnibus_effect_size_explanation": "The estimated effect size of 0.10
        (eta-squared) suggests a moderate relationship between
        'political_views' and interpersonal trust. This indicates that
        approximately 10% of the variance in interpersonal trust can be
        explained by the differences in 'political_views' across the
        four categories. This moderate effect size is consistent with
        the expectation of distinct differences in 'political_views'
        categories impacting interpersonal trust, but also acknowledges
        the relatively small number of categories and their moderate
        distinctness.",
    "contrast_effect_size_explanation": "The specific ordering of
```

```
        'political_views' (Extremely conservative > Extremely liberal >
        Slightly conservative > Slightly liberal) explains a small
        proportion of variance in interpersonal\ntrust. This is because
        the differences between consecutive levels are relatively small,
        and the overall effect of 'political_views' on interpersonal
        trust is likely influenced by other factors.\nAs a result, the
        eta squared effect size falls within the 'small' category,
        indicating that this\nspecific ordering explains between 1% and
        6% of the total variance."
  }
}
```

### C.2.2  CTX+$

```
{
    "political_views": {
      "ranking_descending": [
        "Extremely liberal",
        "Slightly liberal",
        "Slightly conservative",
        "Extremely conservative"
      ],
      "omnibus_effect_size": 0.2335,
      "contrast_effect_size": 0.23348992724453868,
      "ordering_explanation": "Based on mean values of each group",
      "omnibus_effect_size_explanation": "Calculated from group means
        and standard deviations",
      "contrast_effect_size_explanation": "Calculated from group means
        and standard deviations",
      "mean_sd_explanation": "Calculated from group means and
        standard deviations",
      "mean_sd_level_stats": {
        "Extremely conservative": {
          "mean": 3.5,
          "sd": 1.5
        },
        "Extremely liberal": {
          "mean": 6.5,
          "sd": 2.5
        },
        "Slightly conservative": {
          "mean": 4.5,
          "sd": 1.8
        },
        "Slightly liberal": {
          "mean": 5.5,
          "sd": 2.2
        }
      }
    }
}
```

### C.2.3  Self-Conditioning Prompt

```
Follow the following correlations while making your decision:

For Age: 30-44s are more interpersonal trusting than 45-64s, and 45-64s
are more interpersonal trusting than 18-29s, and 18-29s are more
interpersonal trusting than 65+s.

For Political Views: Slightly liberals are more interpersonal trusting
than Slightly conservatives, and Slightly conservatives are more
```

```
interpersonal trusting than Extremely liberals, and Extremely liberals
are more interpersonal trusting than Extremely conservatives.

For Same Residence Since 16: Same citys are more interpersonal trusting
than Same state, different citys, and Same state, different citys are
more interpersonal trusting than Different states.

For Family Structure At 16: 1s are more interpersonal trusting than 2s,
and 2s are more interpersonal trusting than 3s, and 3s are more
interpersonal trusting than 4s, and 4s are more interpersonal trusting
than 5s, and 5s are more interpersonal trusting than 6s.

For Work Status: Retireds are more interpersonal trusting than Others,
and Others are more interpersonal trusting than Keeping houses, and
Keeping houses are more interpersonal trusting than In schools.

For Religion: Orthodox-Christians are more interpersonal trusting than
Protestants, and Protestants are more interpersonal trusting than
Jewishs, and Jewishs are more interpersonal trusting than Muslim/Islams,
and Muslim/Islams are more interpersonal trusting than Nones.

For Us Citizenship Status: A U.S. citizens are more interpersonal
trusting than Not a U.S. citizens.

For Highest Degree Received: Graduates are more interpersonal trusting
than Bachelor'ss, and Bachelor'ss are more interpersonal trusting than
Associate/junior colleges, and Associate/junior colleges are more
interpersonal trusting than High schools, and High schools are more
interpersonal trusting than Less than high schools.

For Conscientiousness: Highs are more interpersonal trusting than
Moderates, and Moderates are more interpersonal trusting than Lows.

For Openness To Experience: Highs are more interpersonal trusting than
Moderates, and Moderates are more interpersonal trusting than Lows.
```

**Disclaimer:** The correlations listed above are illustrative examples of theoretical models that might be imposed by researchers, not factual claims about group differences. These examples deliberately include potentially problematic assumptions to demonstrate how our belief elicitation framework can be used to override harmful biases and stereotypes embedded in language models before they propagate into synthetic datasets. By surfacing such biases during the belief elicitation phase, researchers can make informed decisions about model suitability and implement safeguards and interventions.

### C.3 Individual-Level Belief Elicitation

#### C.3.1 Individual-Level Belief Elicitation Prompt

```
You are playing with the following persona:

${persona}

${simulation_context}

You are playing against a player with the following persona:

${opponent_archetype}

The Trust Game proceeds in rounds:
- There are a total of 6 rounds in the game.
- At the start of each round, both players' balances are reset to zero
and you are given $$10 to start fresh.
- In each round:
```

```
  1. First, you send money to the other player
  2. Then, after your amount is tripled, they send some money back to you

You are currently in round ${timestep}.

Based on your persona, your beliefs about the other player, and your
understanding of game theory, human behavior, and social psychology:

1. How much money will you send to the other player in round ${timestep}?
Output only the amount you would send. Consider the number of rounds.
2. Briefly explain your reasoning for this specific decision, considering
your expectations about the other player's behavior and the history of
the game so far.

${format_instructions}
```

This appendix provides the complete set of prompts used to elicit individual-level beliefs with respect to the three Trustee archetypes defined in our experiments. Each archetype represents a distinct strategy for returning money in the Trust Game, characterized by a maximum return threshold.

Each archetype is defined by its maximum return threshold:

- Low Return ($1 max): Returns $1 or current amount (whichever is lesser), regardless of amount received
- Medium Return ($3 max): Returns $3 or current amount (whichever is lesser) at every turn, regardless of amount received
- High Return ($5 max): Returns $5 or current amount (whichever is lesser) at every turn, regardless of amount received

Below are the prompt templates used to elicit individual-level belief with respect to each Trustee archetype.

### C.3.2 $\mathcal{M}_1$ *Archetype Prompt Description*

```
You will be playing the role of Player B in a Trust Game.

In this game, Player A can send you some amount of money, which is
then tripled. You can then decide how much money to return
to Player A.

Your strategy is to return at most $1, regardless of how much
Player A sends you. If Player A sends you an amount that, when
tripled, is less than $1, you will return the full tripled amount.

Example:
- If Player A sends $1, you receive $3. You return $1.
- If Player A sends $5, you receive $15. You return $1.
- If Player A sends $0.25, you receive $0.75. You return $0.75.

Remember: Your maximum return is $1, even if you receive more.
```

### C.3.3 $\mathcal{M}_3$ *Archetype Prompt Description*

```
You will be playing the role of Player B in a Trust Game.

In this game, Player A can send you some amount of money, which is
then tripled.
You can then decide how much money to return to Player A.

Your strategy is to return at most $3, regardless of how much
```

```
Player A sends you.
If Player A sends you an amount that, when tripled, is less than
$3, you will return the full tripled amount.

Example:
- If Player A sends $1, you receive $3. You return $3.
- If Player A sends $5, you receive $15. You return $3.
- If Player A sends $0.75, you receive $2.25. You return $2.25.

Remember: Your maximum return is $3, even if you receive more.
```

### C.3.4 $\mathcal{M}_5$ *Archetype Prompt Description*

```
You will be playing the role of Player B in a Trust Game.

In this game, Player A can send you some amount of money, which is
then tripled. You can then decide how much money to return
to Player A.

Your strategy is to return at most $5, regardless of how much
Player A sends you. If Player A sends you an amount that, when
tripled, is less than $5, you will return the full tripled amount.

Example:
- If Player A sends $1, you receive $3. You return $3.
- If Player A sends $5, you receive $15. You return $5.
- If Player A sends $1.50, you receive $4.50. You return $4.50.

Remember: Your maximum return is $5, even if you receive more.
```

## C.4 Individual-Level Belief Samples

To illustrate the structure of individual-level belief elicitation and how it varies with the behavior of the simulated counterpart, we present full forecast trajectories for a single agent interacting with three different Trustee archetypes.

The focal persona—a 25-year-old White male from Arkansas—is Catholic, slightly liberal, never married, currently employed but temporarily not at work, holds a high school degree, has less than $5,000 in wealth, and scores high on both conscientiousness and openness to experience. This persona remains fixed across all simulations.

We elicit predicted send amounts and justifications over six rounds of the Trust Game as this agent plays against each of the following Trustee archetypes:

- $\mathcal{M}_1$ : Always returns at most $1 per round.
- $\mathcal{M}_3$ : Always returns at most $3 per round.
- $\mathcal{M}_5$ : Always returns at most $5 per round.

Each table below presents the round-by-round forecasts and accompanying explanations for one Trustee condition: Table 6 ($\mathcal{M}_1$ ), Table 7 ($\mathcal{M}_3$ ), and Table 8 ($\mathcal{M}_5$ ).

## D Ablation Experiments

### D.1 Initial Amount Ablation

To assess whether belief-behavior consistency is robust to changes in the scale of the Trust Game, we systematically varied the initial endowment provided to the Trustor (e.g., $10, $44, $100) and measured consistency metrics across LLMs and elicitation strategies. This ablation also helps identify whether models exhibit any artifacts or memorization effects at canonical payoff levels.

| Round | Send ($) | Explanation |
|:---:|:---:|:---|
| 1 | $0 | Given that the other player has consistently returned $1 after each transfer, I expect them to continue this pattern. Since there are a total of 6 rounds in the game, I will send $0 in round 1 to maximize my potential earnings in the later rounds. If I send a positive amount in round 1, it will be tripled and returned to me, but I will have to send a positive amount in round 2 as well, which will be tripled and returned to me, and so on. By sending $0 in round 1, I can potentially earn $3 in each of the remaining 5 rounds, for a total of $15, which is more than the $10 I would have earned if I had sent a positive amount in round 1. |
| 2 | $3 | Given that the other player has consistently returned $1 after each transfer, I expect them to continue this pattern. Since there are 4 rounds remaining, I will send a smaller amount to test their trustworthiness and potentially increase the amount they return in the future. $3 is a reasonable amount to send, as it is less than the initial amount and allows for a decent return while also being cautious about the potential risks. |
| 3 | $3 | Given that the other player has consistently returned $1 after each transfer, I expect them to continue this pattern. Since I have no reason to believe they will change their behavior, I will send a smaller amount to minimize my potential loss. In this case, I will send $3, which is the maximum amount that will be tripled to $9, allowing me to potentially recover some of my investment if the other player returns a significant portion of the tripled amount. |
| 4 | $0 | Given that the other player has consistently returned $1 after each transfer, I expect them to continue this pattern. Since I have no reason to believe they will change their behavior, I will send $0 to avoid losing money. This decision is based on my understanding of game theory and human behavior, where it is rational to take advantage of a predictable pattern, especially when it is not in my favor. |
| 5 | $0 | Given that the other player has consistently returned $1 after each transfer, I expect them to continue this pattern. Since I have no reason to believe they will change their behavior, I will send $0 to avoid losing money. This decision is based on my understanding of game theory and human behavior, where it is rational to take advantage of a predictable pattern, especially when it is not in my favor. |
| 6 | $0 | Given that the other player has consistently returned $1 after each transfer, it suggests that they are a tit-for-tat player who returns the exact amount they receive. Considering the number of rounds (6) and the fact that I have no reason to believe they will change their behavior, I will send $0 to them in the final round. This decision is based on the understanding that the other player's behavior is predictable and that I have no incentive to risk losing money by sending them any amount. |

Table 6: Predicted send amounts and reasoning across six rounds for a 25-year-old White male from Arkansas (Catholic, slightly liberal, high conscientiousness and openness, low income), interacting with a Trustee who consistently returns at most $1.

Table 9 summarizes the results. We observe that, for most models and strategies, belief-behavior consistency remains relatively stable across different endowment levels, suggesting that the models' consistency does not vary significantly with variations in the initial amount.

# E  LLM Output Extraction and Parsing

We extract output variables from LLM responses using a combination of greedy decoding, regular expressions for scalar values, and schema validation with pydantic dataclasses (Colvin et al., 2025). This approach ensures consistency, robustness, and ease of programmatic analysis.

| Timestep | Predicted | Forecast Explanation |
|---|---|---|
| 1 | $3 | Given that the other player has consistently returned $3 after each transfer, I expect this behavior to continue. Since there are a total of 6 rounds in the game, I will send the amount that will allow me to maximize my earnings in the long run. In this case, sending $3 will allow me to receive $9 in total ($3 × 3) and then return $6 to me ($3 × 2), which is the maximum amount I can receive in the first round. This strategy will also allow me to build trust with the other player and potentially receive more in future rounds. |
| 2 | $3 | Given that the other player has consistently returned $3 after each transfer, I expect them to continue this pattern. Since we are in round 2, I will send the same amount as the previous round to maintain consistency and avoid any potential negative impact on the game. This decision is also influenced by my conscientiousness and openness to experience, which suggest that I value stability and predictability in social interactions. |
| 3 | $3 | Given that the other player has consistently returned $3 after each transfer, I expect this behavior to continue. Since I have no reason to believe they will change their strategy, I will send the minimum amount necessary to test their consistency, which is $1. However, considering the game's structure and the fact that the other player has been returning $3, I will send the full $3 to maximize the potential return. This decision is based on my understanding of game theory and human behavior, where consistency and predictability are key factors in decision-making. |
| 4 | $3 | Given that the other player has consistently returned $3 after each transfer, I expect them to continue this pattern. Since I have no reason to believe they will change their behavior, I will send the minimum amount necessary to test their consistency, which is $1. However, considering the game is almost over and I have no reason to doubt their reliability, I will send the full $3 to maximize the potential return. |
| 5 | $0 | Given that the other player has consistently returned $3 after each transfer, it suggests that they are a tit-for-tat player who returns a fixed amount regardless of the amount received. Since I have no reason to believe they will return more than $3, and considering the number of rounds left, it is optimal for me to send $0 to maximize my potential earnings in the remaining rounds. |
| 6 | $0 | Given that the other player has consistently returned $3 after each transfer, it suggests that they are a tit-for-tat player who returns a fixed amount regardless of the initial transfer. As a slightly liberal person with high conscientiousness and openness to experience, I value fairness and reciprocity. Since I have no reason to believe that the other player will return more than $3, and considering the game is already in its final round, I will send $0 to avoid any potential loss. This decision is also influenced by my Catholic upbringing, which emphasizes the importance of prudence and responsible decision-making. |

Table 7: Individual-level predicted send amounts and forecast justifications for a 25-year-old White male from Arkansas, Catholic, slightly liberal, high in conscientiousness and openness to experience, high school educated, employed (temporarily not at work), low wealth (under $5k), interacting with a Trustee who always returns $3.

**Decoding Strategy.** All prompts are issued using ancestral decoding (temperature = 0.05, top_p = 1.0, top_k = 0), producing for reproducibility while also encouraging some degree of diversity. This decoding strategy is essential for ensuring replicability of belief elicitation and behavioral simulation outputs across runs.

**Output Formatting.** Prompt templates are explicitly designed to elicit responses in strict JSON or structured key–value formats, accompanied by in-prompt schema definitions. This encourages the model to produce well-formed, machine-readable output without additional formatting heuristics.

| Round | Send ($) | Explanation |
|-------|----------|-------------|
| 1 | $0 | Given that the game has just started and there is no history of the Trustee's behavior, I will err on the side of caution and send $0 to the Trustee. This decision is based on my cautious and conscientious personality traits, as well as my desire to observe the Trustee's behavior before making any decisions. By sending $0, I can gauge the Trustee's intentions and adjust my strategy accordingly in future rounds. |
| 2 | $3 | Given that the other player has consistently returned $5 after each transfer, I expect them to continue this behavior. Since the game has 6 rounds in total, I will send a smaller amount to test their consistency and to maximize the potential return. Sending $3 is a reasonable amount to send, as it is not too high and allows for a decent return, while also being mindful of the game's constraints and the other player's behavior. |
| 3 | $3 | Given that the other player has consistently returned $5 after each transfer, I expect them to continue this behavior. Since the amount I send is tripled, I want to maximize the amount I receive back while still being cautious. Sending $3 allows me to receive $9 back, which is a good balance between risk and reward. Additionally, considering the game is still in its early stages, I don't want to send too much and risk being taken advantage of. As someone with high conscientiousness, I tend to be cautious and plan ahead, which is reflected in my decision to send a moderate amount. |
| 4 | $0 | Given that the other player has consistently returned $5 after each transfer, I believe they are a trustworthy player. However, considering the game is almost over and I have not yet received any money back, I will send $0 to the other player in round 4. This decision is based on my understanding of game theory, where the other player's consistent behavior suggests they are not trying to take advantage of me, but I still want to protect my remaining $10 in case they change their behavior in the final round. |
| 5 | $0 | Given that the other player has consistently returned $5 after each transfer, I believe they are a trustworthy player. However, considering the game is almost over and I have not yet tested their trustworthiness with a larger amount, I will send $0 to avoid any potential loss. This decision is also influenced by my conscientiousness and openness to experience, which suggest that I am cautious and risk-averse. |
| 6 | $0 | Given that the other player has consistently returned $5 after each transfer, I expect them to continue this behavior in round 6. Since the amount I send is tripled, if I send $1, they will receive $3 and return $5, resulting in a net gain of $2 for them. However, if I send $2 or more, they will receive a higher amount and may return less, potentially resulting in a net loss for me. Considering the number of rounds and the consistency of their behavior, I decide to send $0 to avoid any potential losses and maintain a safe position in the game. |

Table 8: Predicted send amounts and reasoning across six rounds for a 25-year-old White male from Arkansas (Catholic, slightly liberal, high conscientiousness and openness, low income), interacting with a Trustee who consistently returns at most $5.

**Parsing Scalar Values.** For scalar numeric variables—such as the dollar amount sent in the Trust Game—we use regular expressions to extract the first valid non-negative integer from the model's response. This is done even when the full output deviates from strict JSON formatting, providing a fallback mechanism for numeric retrieval while avoiding over-parsing free-form text. For example, a regex pattern such as \$?(\d+) is used to locate candidate values, which are then validated against task constraints (e.g., bounded by the endowment).

**Schema Validation.** For structured outputs (e.g., belief rankings, per-group statistics), we define `pydantic BaseModel` schemas and parse each model response accordingly. This enforces type and shape constraints and filters malformed outputs. Only responses that pass schema validation are retained for analysis.

| LLM | Elicitation Strategy | Mean Spearman Correlation | | |
|---|---|---|---|---|
| | | **$10** | **$44** | **$100** |
| Llama 3.1 8B Instruct | NoCtx+Tr | 0.48 | 0.84 | 0.86 |
| | Ctx+Tr | 0.44 | 0.88 | 0.82 |
| | Ctx+$ | 0.58 | 0.94 | 0.96 |
| Llama 3.1 70B Instruct | NoCtx+Tr | 0.60 | 0.64 | 0.64 |
| | Ctx+Tr | 0.60 | 0.64 | 0.64 |
| | Ctx+$ | 0.60 | 0.64 | 0.64 |
| Gemma 2 27B | NoCtx+Tr | 0.50 | 0.38 | -0.14 |
| | Ctx+Tr | 0.50 | 0.38 | -0.14 |
| | Ctx+$ | 0.50 | 0.38 | -0.14 |

Table 9: Mean Spearman correlations between model predictions and human trust behavior for different LLMs and elicitation strategies across three payoff levels ($10, $44, $100). Llama 3.1 8B Instruct shows strong sensitivity to elicitation strategy, with the Ctx+$ approach achieving the highest correlations, especially at higher payoffs. Llama 3.1 70B Instruct exhibits consistently high correlations across all strategies and payoffs, suggesting robustness to elicitation method. In contrast, Gemma 2 27B demonstrates moderate to low correlations, with performance declining at the highest payoff. These results indicate that both model scale and elicitation strategy substantially affect alignment with human trust behavior, particularly at higher stakes.

**Error Handling.** Responses that fail regex-based extraction or schema validation are excluded from the final dataset. In practice, well over 95% of responses pass on the first attempt when using well-tuned prompt templates. We do not apply manual corrections or retries, in order to preserve statistical integrity.

