# OpenReview forum: "Do Role-Playing Agents Practice What They Preach? Belief-Behavior Alignment in LLM-Based Simulations of Human Trust"
_colmweb.org/COLM/2025/Workshop/Social_Sim — Social Sim'25_

### Official Review · Reviewer_2jZ7 · 2025-07-16

**Rating:** 7
**Overall Assessment:** 3
**Confidence:** 4

**Review:**

See above.

**Comments Suggestions And Typos:**

see above.

**Ethical Concerns:**

no ethical concerns.

**Paper Summary:**

The paper investigates the consistency between LLM-based role-playing agents' stated beliefs and their simulated behaviors, using the Trust Game as a testbed. The authors introduce a framework to measure belief-behavior consistency at both population and individual levels, exploring factors like elicitation strategies, task context, and forecasting horizons. Key findings reveal systematic inconsistencies, with self-conditioning improving alignment in some models but imposed priors often undermining it.

**Relevance:**

5

**Summary Of Strengths:**

The proposed framework for assessing belief-behavior consistency is well-designed and addresses a critical gap in evaluating LLM-based role-playing agents.

The topic is timely and important, given the increasing use of LLMs for synthetic data generation in behavioral research.

**Summary Of Weaknesses:**

The workload appears modest, with experiments primarily focused on the Trust Game and a relatively small set of personas. However, it should be sufficient for the workshop.

---

### Official Review · Reviewer_GRg2 · 2025-07-17

**Rating:** 9
**Overall Assessment:** 4
**Confidence:** 5

**Review:**

Overall, this paper is well-motivated and addresses something that have not really been touched upon by the social sims community. The paper is overall clearly written. There are some minor questions in terms of making sense of results but overall this is quite good.

**Comments Suggestions And Typos:**

N/A

**Paper Summary:**

This paper purposes to evaluate the degree of consistency between role-playing LLMs’ stated belief vs. actual behavior when tasked with role-playing simulations. Specifically they study the well-known Trust Gane and measure to what degree 1) at population level, LLMs are consistent in terms of belief-behavior 2) at individual level, how well could LLMs predict their own actions.

**Relevance:**

5

**Summary Of Strengths:**

Novel angle, good execution, interesting result.

**Summary Of Weaknesses:**

1.	In terms of operationalization: The current method for belief elicitation relies on prompting the LLM to perform an analytical task (e.g., "act as a researcher and rank these attributes"). This raises the possibility that the observed inconsistency is not a failure of a unified agent's "belief-behavior" link, but rather an artifact of the LLM "code-switching" between two distinct task modes: an analytical mode for elicitation and a narrative/role-playing mode for simulation. The argument would be substantially strengthened by exploring alternative belief elicitation methods, such as probing intermediate model representations (e.g., using techniques like logic lens), to see if a more "native" form of belief yields different results.

2.	Belief vs. behavior: I am not sure what to expect LLMs to have – I mean there are plenty of theories for humans “value-action” gap but LLMs are no humans and don’t have the same underlying mechanism as humans so I am not sure whether any levels of belief-action consistency is meaningful? Additionally, do you have a sense of how much consistency humans have on the particular settings this paper use?

---

### Meta-Review · Program_Chairs · 2025-07-24

**Recommendation:** Accept

**Metareview:**

--